# The first luminescence dating of Tibetan glacier basal sediment

Zhu Zhang, Shugui Hou, Shuangwen Yi

School of Geographic and Oceanographic Sciences, Nanjing University, Nanjing, 210093, China

*Correspondence to*: Shugui Hou (shugui@nju.edu.cn)

**Abstract.** Dating of ice cores drilled in the high mountain glaciers is difficult because seasonal variations cannot be traced at depth due to rapid thinning of the ice layers. Here we provide the first luminescence dating of the basal sediment of the Chongce ice cap in the northwest Tibetan Plateau. Assuming the sediment is of similar (or older) age as the surrounding ice, the luminescence dating of 42 ± 4 ka provides an upper constraint for the age of the bottom ice at the drilling site. This result is more than one order of magnitude younger than the previously

suggested age of the basal ice of the nearby Guliya ice cap (~40 km in distance).

## 1 Introduction

Ice cores from the high elevation regions provide a wealth of information for past climatic and environmental conditions that extends beyond the instrumental period. A precise chronology is the essential first step for a reliable

interpretation of the ice core records. The Tibetan Plateau (TP), sometimes called "the Roof of the World", is the world's highest and largest plateau with an average elevation exceeding 4000 m above sea level (a.s.l.) and an area of 2.5 Million square kilometers. It has the largest number of glaciers outside the polar regions. In 1992, a 308.6 m ice core to bedrock was recovered from the Guliya ice cap on the northwest TP (Fig. 1). Top 266 m of the core was dated to a period spanning 110 ka, and the ice below 290 m was suggested to be more than 500 ka old due to $^{36}$Cl-

dead in the ice (Thompson et al., 1997). This makes it the second oldest continuous ice core, only younger than the Antarctic EPICA Dome C ice core. During the past two decades, the Guliya ice core record has been widely used as a benchmark for correlating regional climate variables in the Westerlies region of the central Asia and the northern Tibetan Plateau. However, Cheng et al. (2012) suggested a substantial revision of the Guliya chronology due to the enormous inconsistence between the $\delta^{18}$O records of the Guliya ice core and the Kesang stalagmite (Fig. 1 and

Supplement).

Luminescence dating can be readily applied to most terrestrial sediments and can be used to date sediments on timescales from $10^1$ to $10^5$ years, encompassing the entire late Quaternary (Fuchs and Owen, 2008). Dating errors within a few percent of the age can be achieved depending upon the nature of the sediment and the laboratory

methods. During the recent decades, optically stimulated luminescence (OSL) method has been successfully used for dating glacial sediments on the TP and surrounding regions (e.g., Owen et al., 2003; Ou et al., 2014; Hu et al., 2015). Willerslev et al. (2007) provided the first luminescence measurements on the single grains of quartz and feldspar extracted from a sample cut out of an opaque part of the Greenland Dye 3 basal ice containing dispersed sandy and silty particles. To our knowledge, this is the only published luminescence dating of glacier basal ice so far. As pointed out by the authors, at the time when the Dye 3 ice core was drilled (1979–1981), no standard apparatus or procedures were used to avoid inadvertent exposure to light during the drilling and subsequent handling, inspection and storage of the ice cores. For this reason, the sample may be considered to be highly problematic for luminescence dating (Willerslev et al., 2007).

## 2 The Chongce ice cores

The Chongce ice cap is located in the west Kunlun Mountains on the northwest TP, with a snowline altitude about 5900 m a.s.l.. It is 28.7 km in length, covering an area of 163.06 $km^2$, with a volume of 38.16 $km^3$. In 2012, we drilled two ice cores to the bottom with length of 133.8 m and 135.8 m respectively, and a shallow ice core with length of 58.8 m at an altitude of 6010 m a.s.l. from the Chongce ice cap (Fig. S1). The measured borehole temperature is about -8.8 °C at depth of 130 m. The very bottom of the core is a mixture of sediment and ice (see picture in Fig. S1). No special precautions were taken to avoid exposure to light during the drilling operation, so this mixture is not suitable for luminescence dating. Therefore, two more ice cores to the bottom were recovered with length of 216.6 m and 208.6 m respectively at an altitude of 6100 m a.s.l. (35°14'57" N, 81°5'28" E. Fig. S1) in 2013. This time we managed to avoid exposure to light during the drilling and subsequent procedures. The drilling was performed at night. When it was close to the bottom, the cores, together with chips, were recovered in faint red light. Afterwards, the cores and chips were directly sacked into opaque plastic bags and immediately wrapped with aluminium foil and adhesive tape. The ice cores were kept frozen and transported to the cold room in the Nanjing University for further processing.

## 3 Sample preparation and measurements

Sample preparation was performed under the safe red light in the Luminescence Dating Laboratory of the Nanjing University. The total bottom sediment including ice is 1431.7 g. We first took a small portion of the sediment (13.4 g) for measuring ice content, which is determined by weighting the mass before and after drying, resulting in ~30%

ice (water equivalent) content. Therefore, we assume that there might be ~30% water at the bottom of the ice core drilling site during the warm period(s) in history. Moreover, because the water content may have varied over the entire burial period of the sample, we assigned a relative uncertainty of 50% to allow for possible fluctuations. Afterwards, this part of sediment was dried in oven (<60 °C). About 5 g dried sediment was ground to fine powder

for determining U, Th and K concentrations by neutron activation analysis (NAA) at the China Institute of Atomic Energy, Beijing. The results are given in Table 1. As the measured concentrations are normally low, the accuracy of the dose rates, calculated from the NAA results, may have been affected by the inhomogeneity of the deposits. Therefore, ~ 100g sample was also measured for >24 h using the high-resolution gamma spectrometry (Table S1). The sample was ignited at 450 °C, cast in wax and stored for >3 weeks to assure equilibrium conditions for $^{226}$Ra–

$^{222}$Rn decay (Murray et al., 1987).

The rest of the sediment (~1300 g) was put into a beaker and melted in the Luminescence Dating Laboratory at room temperature (~ 20 °C). The melt water was carefully extravasated. This process was repeated to leave as little water inside the beaker as possible. Then the sediment was filtered through the 200 μm wet sieve. The filtered sediment was successively soaked in the 10% hydrochloric acid and 30% hydrogen peroxide to remove carbonate

and organic matters. Afterwards, the residual was wet-sieved to separate the coarse grains (90 - 150 μm) from the fine grains (<40 μm).

The coarse grains were then separated by dense liquid (2.58 g.cm$^{-3}$), resulting in the coarse quartz grains at the bottom. After purified with deionized water, the coarse (90 - 150 μm) quartz grains were immersed in 40% hydrofluoric acid for 40 minutes to remove any remaining feldspar contamination, and in 10% hydrochloric acid for

40 minutes to remove residual fluoride. The purity of the quartz was determined using the OSL-IR depletion ratio method (Duller, 2003). Even after being etched (40% HF, 40 minutes) twice, it was not possible to remove feldspar completely. So, a post-IR measurement with blue LED stimulation (Banerjee et al., 2001) was applied to the coarse grain quartz measurements.

As for the fine grains (<40 μm), a fraction (4 - 11 μm) was further isolated according to Stokes' law. To extract pure

fine quartz grains, the sample was submerged first in the 40% fluosilicic acid for 10 days, and then in the 10% hydrochloric acid to remove any fluorides. The purity of treated fine grains was checked by the same method as that used with the coarse grains quartz (Duller, 2003). Samples with IRSL (infrared stimulated luminescence) vs. blue OSL signal ratios over 10% of unity would be re-treated with fluosilicic acid until the ratio was within 10% of unity. The coarse quartz grains (90 - 150 μm) were then mounted on stainless steel discs (~2 mm in diameter) using

the Silkospray silicone oil, and the fine quartz grains (4 - 11 μm) were settled on discs using pure water with a pipette. Measurements were performed on a Risø TL/OSL reader (model DA-20C/D), fitted with blue LEDs (470 nm, ~80 mW.cm$^{-2}$), infrared LEDs (870 nm, ~135 mW.cm$^{-2}$) and a $^{90}$Sr/$^{90}$Y beta source (Bøtter-Jensen et al., 2010).

The OSL signal from the quartz grains was detected by a 9235QA photomultiplier tube through a 7.5 mm Schott U-340 detection filter.

The quartz equivalent doses ($D_e$) were determined using the single aliquot regeneration (SAR) protocol (Murray and Wintle, 2000). A preheat temperature of 260 °C with a cutheat of 220 °C was selected for $D_e$ measurement with an elevated temperature (280 ˚C) blue light bleach at the end of each cycle. The early background subtraction (first 0.16 s minus background from 0.16 - 0.32 s interval) was used for signal integration, and to minimize the influence of slow and medium components (Cunningham and Wallinga, 2010).

Individual $D_e$ values were obtained using a single-saturating exponential fitted in Analyst version 4.31.7 (Duller, 2015). The uncertainty of individual $D_e$ values was calculated using counting statistics and an instrumental uncertainty of 1.5%.

## 4 Results and discussion

The basic assumption of the OSL dating technique is that the mineral grains were fully bleached before deposited. The possibility of complete bleaching of the grains decreases for materials from supraglacial debris, to englacial debris, to basal debris (Fuchs and Owen, 2008), suggesting that extreme care should be taken when applying the luminescence dating techniques to basal debris. The OSL dose-response and decay curves of the quartz aliquots are shown in Fig. 2. It's apparent that parts of the coarse grain (90 - 150 μm) quartz aliquots approach the saturated level, but all the fine grain aliquots are similar and well below the saturated level. The decay curves of the fine grain (4 - 11 μm) show that the OSL signal decreases very quickly and approaches to background level during the first seconds of stimulation. We further decomposed the natural OSL signal of the fine grain aliquots into fast, medium and slow components (Fig. S2). Fast component accounts for around 87% of the natural signal, indicating that the signal is fast component dominant. However, the OSL decay curves of the coarse grain aliquots display a relatively slow trend (Fig. 2). We also performed scanning electron microscope of the sediment sample. A typical coarse grain quartz is shown in Fig. S3. Its angular texture and low degree of deformation suggest that this coarse grain quartz might have been scoured from bedrock, which mainly consists of Triassic high-K granitoids in the west Kunlun Mountains (Wang et al., 2013). In addition, the $D_e$ distribution presented in Fig. 3 shows a narrow and nearly symmetric shape for the fine quartz grains, possibly indicating sufficient bleaching of the fine quartz grains. These results imply that the fine grain quartz might be more appropriate for luminescence dating than the coarse grain quartz. The same conclusion was also reached in some previous studies on Himalayan glacial deposits (e.g. Hu et al. 2015). We therefore used the results of the fine grains, and the results from the coarse grain quartz were given in the Supplement (Table S3). Although the coarse grain ages are slightly older than the fine grain ages (Table 1), they clearly fall within a similar range.

We further evaluate the reliability of the quartz grain measurements by their recuperation and recycling ratios (Table S2). The recuperation values of all the fine grain quartz aliquots are below 5% (Fig. S4). The recycling ratios range from 0.82 to 1.2, with 9 of the 12 fine grain quartz aliquots having recycling ratios within the acceptable range of 0.9 to 1.1. Only the aliquots with recycling ratios within the acceptable range were used for

further analysis. Among these, one additional aliquot was also excluded because its $D_e$ value fell outside 2σ of the distribution (Fig. 3). In the end, 8 fine grain quartz aliquots were used for OSL age calculation, resulting in an average equivalent dose ($D_e$) of 178 ± 9 Gy (Table 1).

Annual dose rate (D) can be estimated from the U, Th and K concentrations, water contents and cosmic ray contribution (Guérin et al., 2011). An alpha efficiency factor (a-value) of 0.04 ± 0.02 for quartz (Rees-Jones, 1995)

was used to calculate the alpha contribution to the total dose rate. The cosmic dose rate is negligible because the sediments are covered by thick ice and multiple dust layers, hence largely insulated from light and cosmic rays. There are potentially two additional sources of radiation that are not included in the dose rate calculation: (1) radiation from the bedrock, and (2) radiation from dust layers in the core. Willerslev et al. (2007) suggested that the dose rate contribution from the underlying bedrock at the Greenland Dye 3 ice core drilling site was negligible

following calculations based on radiation transport modelling software (MCNP5). Because no literature values are available to calculate the dose rate contribution from the underlying bedrock at our drilling site, at the first stage it is assumed that its contribution to the dose rate was insignificant. As to radiation from dust layers in the core, we have not measured the dust concentrations of our Chongce ice cores, but A 18.7 m ice core drilled at the summit (6530 m a.s.l.) of the Chongce ice cap in 1992 gives a maximum dust mass concentration of 955 mg.kg$^{-1}$ (Li et al.,

2006). This provides a general impression of dust layers along the Chongce ice cores, which is ~3 orders of magnitude lower than the bottom sediment, suggesting insignificant influence of radiation from dust layers in the core, given a similar radiation intensity of the dust layers in the core and the bottom sediment. Therefore, increased dose rate would be expected if these additional sources of radiation were included, thus resulting in a younger age. The converted U, Th and K concentrations from the gamma measurements are close to the NAA results (Table 1),

affirming the accuracy our element measurements. It's well recognized that water content is one of the major uncertainty factors of the Luminescence dose rate calculation because the real situation of sediments during geological history is not known. To approach this, two extreme situations for the sediment were presumed, one is zero water content which represents no ice (or water), and the other is the sediment full immersed by the water (saturation water content), i.e., 30 ± 15% water content in our case. Under these two situations, the maximum and

minimum dose rates from water content were estimated. For these uncertainties, only an upper age constraint of 42 ± 4 ka is given in Table 1.

The dated fine grain quartz aliquots are a mixture of eolian quartz and subglacially scoured quartz. Sharp and Gomez (1986) suggested that quartz particles should be "tools" producing comminution of softer minerals within subglacial environments while they themselves remain unaltered. Wright (1995) performed glacial grinding simulations by using a ring shear box designed to examine abrasion and size reduction of quartz grains under

"grinding" conditions analogous to those experienced in a subglacial environment. The findings from experimental runs suggest that, although particle breakage and comminution does occur, little silt-sized material is actually produced. The amount of silt produced ranges from approximately 9% of the original sample in one experimental run to less than 1% for the other seven runs. The direct evidence is from Fig. S2, indicating that fast component of the fine grain aliquots accounts for around 87% of the natural OSL signal. So the signal is fast component dominant.

If for a high bedrock fraction, medium and slow components should be high due to no or less bleaching of the bedrock. So the fine quartz grains used for the dating are mostly an eolian origin, implying that the luminescence dating would represent the time for the ice to move from the surface to the bed.

Our new data does not imply for an ice-free region in the Chongce region since $42 \pm 4$ ka ago, but for an ice-free condition below the elevation of the bottom at the drilling site during a (or more) warm period (or periods) since

this upper constraint age (e.g., the Bølling-Allerød period, Holocene Climate Optimum). Takeuchi et al. (2014) reported radiocarbon dating of organic soil from the bottom of an 86.87 m ice core drilled at the top of the Grigoriev Ice Cap (41°58'33" N, 77°54'48" E) in the Tien Shan Mountains, showing that the age of the soil is 12 656 - 12 434 cal. years before present. Therefore, Takeuchi et al. (2014) suggested that the Grigoriev Ice Cap did not exist in the Bølling-Allerød period, which gives support to our results.

Given the surface elevation of the Chongce drilling site of 6100 m a.s.l. and the ice core length of 216.6 m, the elevation at the bottom of the Chongce drilling site is 5883.4 m a.s.l.. As to the Guliya ice core drilling site, its surface elevation of 6200 m a.s.l. and its length of 308.6 m result in an elevation at the bottom to be 5891.4 m a.s.l., suggesting that the age of the bottom ice at the Chongce and the Guliya drilling sites might be comparable. Thus our new data can not support the previously suggested age of more than 500 ka old of the Guliya bottom ice

(Thompson et al., 1997).

**5 Conclusions**

An upper age constraint ($42 \pm 4$ ka) of the basal sediment sample from the Chongce ice cap in the west Kunlun Mountains on the northwest TP is estimated, which is more than one order of magnitude younger than the previously suggested age of bottom ice in the west Kunlun Mountains. The major limitation of the current work is

the very small number of absolute ages and simple assumptions, and there exists much uncertainty, such as determination of the dose rate, additional radiation sources from bedrock and dust layers in the core, specific

processes underneath the glacier, and so on. For these reasons, we only suggest an upper constraint for the age of the sediment. Future work should include collecting more glacier basal sediment samples for the luminescence dating, better understanding the unique processes for preserving the luminescence signal in the glacier basal sediment, and effect of ice on for the dose rate. However, the current work provides potential implications for exploring age of mountainous glacier bottom ice, and an important step towards better understanding the Tibetan ice cores and more accurate interpretation of their records.

**Author contribution**

Shugui Hou designed this work and drilled the ice cores. Zhu Zhang and Shuangwen Yi performed the measurements. Shugui Hou wrote the paper. All authors contributed to discussion of the results.

**Competing interests**

The authors declare that they have no conflict of interest.

**Acknowledgments**

Thanks are due to many scientists, technicians, graduate students and porters for their hard work in the field, to Hongxi Pang and Wangbin Zhang for help in sampling, and to Zhongpin Lai, Lupeng Yu and Yuanfeng Cai for discussion. This work was supported by the National Natural Science Foundation of China (41330526). We are grateful to three anonymous reviewers, Joel Gombiner and the editor Julia Boike for their constructive comments, which helped to improve the paper.

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

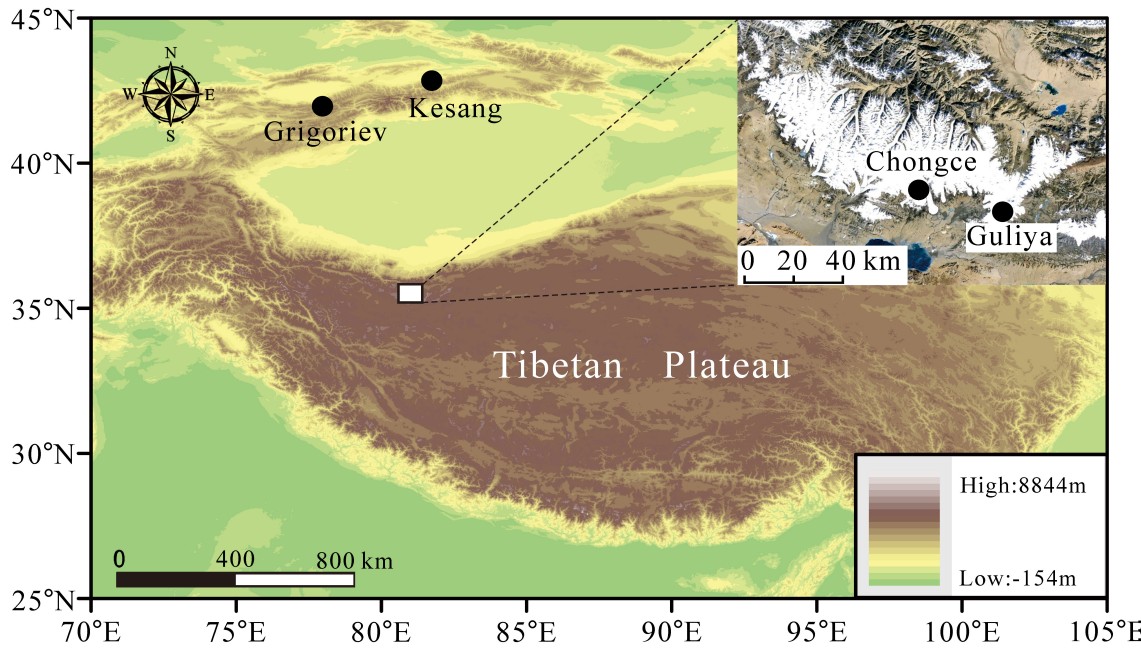

Figure 1: Study area map showing the locations of the Chongce and Guliya ice caps, the Kesang cave and the

5    Grigoriev ice cap.

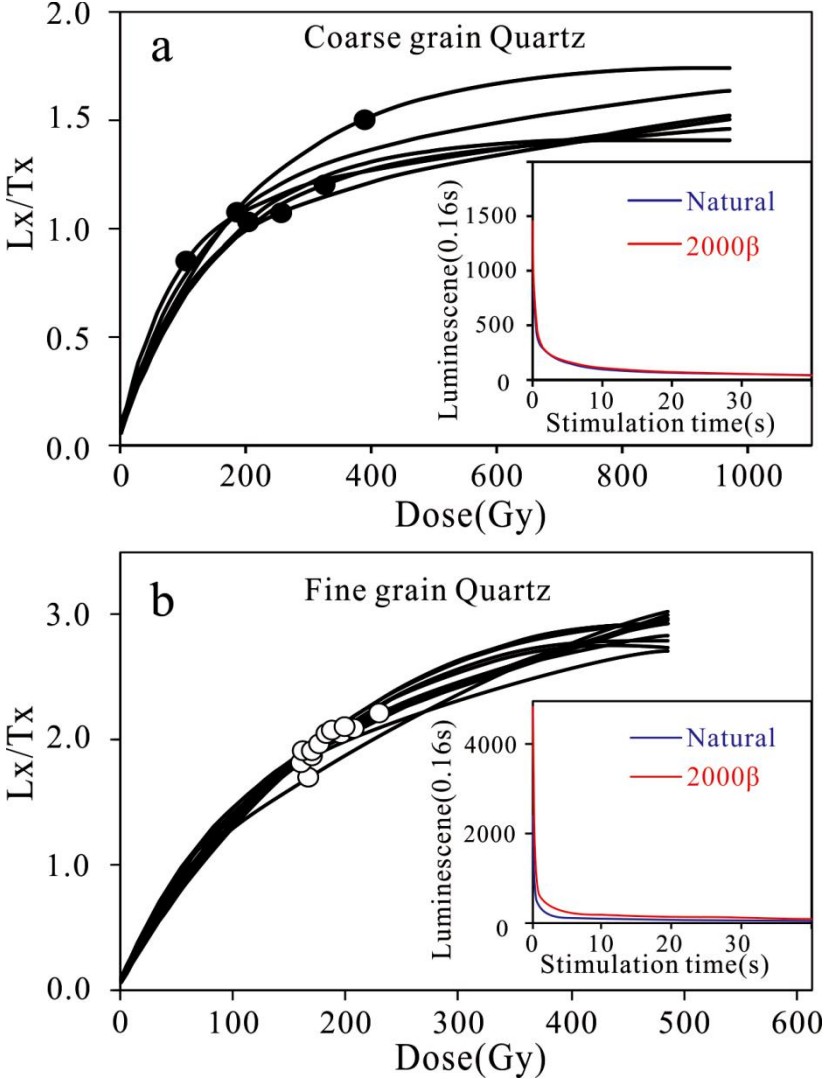

Figure 2: The OSL dose-response and decay curves of the coarse and fine grain quartz aliquots. Inserts show decay curves of the natural and regenerated dose (2000β) of the sample.

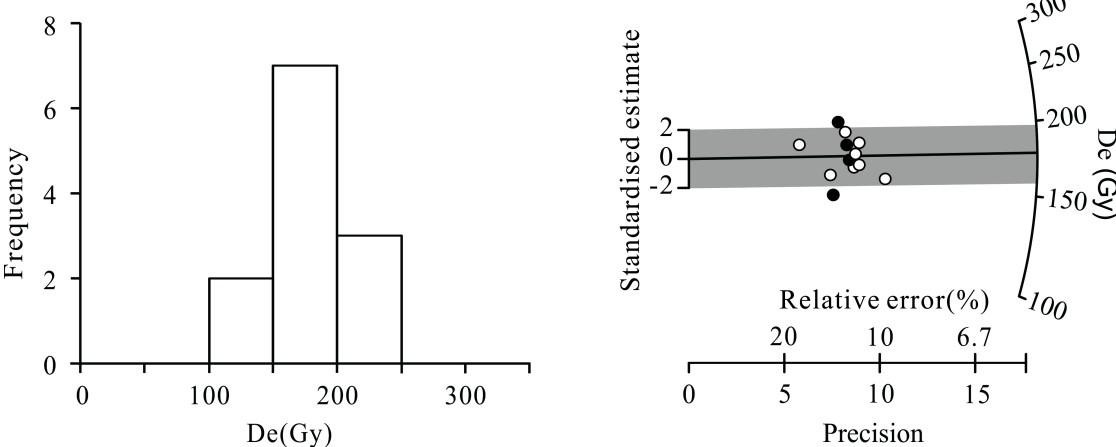

Figure 3: $D_e$ distribution histogram (left) and radial plot (right) showing the distribution of $D_e$ results for the fine grains. The shaded region represents $2\sigma$ width of the distribution. The filled symbol represents the aliquots that were excluded in the final age calculation.

Table 1. Results of the fine quartz grains with their corresponding OSL ages.

| Sample | U (ppm) | | Th (ppm) | | K (%) | | $D_e$ | Water content | Dose rate | Age |
|---|---|---|---|---|---|---|---|---|---|---|
| | Gamma | NAA | Gamma | NAA | Gamma | NAA | (Gy) | (%) | (Gy/ka) | (ka) |
| CCICE* | 3.66 ± 0.15 | 3.45 ± 0.12 | 11.21 ± 0.42 | 11.40 ± 0.32 | 3.52 ± 0.10 | 3.48 ± 0.08 | 178 ± 9 | 0 | 5.81 ± 0.46 | 31 ± 3 |
| | | | | | | | | 30 ± 15 | 4.24 ± 0.37 | 42 ± 4 |

*CCICE stands for Chongce Ice.