# Peer review of "The first luminescence dating of Tibetan glacier basal sediment"

_The Cryosphere, 2017_

## Short Comment (SC1) · 17 May 2017

This is an interesting paper that should eventually be published.

However, the authors could give more thought to the calculation of dose rate and the meaning of the optical age for basal sediment.

The OSL age is the equivalent dose divided by the dose rate. In calculating the OSL age, the authors calculated a lower limit age for dehydrated sediment, containing air in the pore space, and an upper limit age, for hydrated sediment containing water in the pore space. The dose rate is lower for hydrated sediment because water attenuates radiation transfer from grain to grain. The actual sample came from sediment embedded within ice. The authors should calculate a dose rate for the real situation of sediment in ice.

[Figure]

There are potentially two additional sources of radiation that are not included in the dose rate calculation.

(1) Radiation from the bedrock or subglacial sediment.

(2) Radiation from dust layers in the core.

The authors should add these sources of radiation to the dose rate, or show that they are insignificant. If these other sources are included, the higher radiation dose rate would lower the calculated age.

Finally, I am not sure that the OSL age of the basal sediment directly relates to the age of the ice cap. The authors suggest that the sand-sized quartz grains are sourced from subglacial erosion. If true, it seems likely that some of the silt-sized quartz is also derived from subglacial erosion. Thus, it is conceivable that the dated aliquots are a mixture of eolian quartz and subglacially derived quartz.

If the ice flow at the core site is dominated by downward vertical motion, then the OSL age of the eolian component of the dated aliquots would represent the time for the ice to move from the surface to the bed, not the age of the ice cap itself.

---

## Referee Comment (RC1) · Anonymous Referee #1 · 23 May 2017

The manuscript by Zhang et al. provides interesting and new data which justify publication in The Cryosphere. It is relatively well structured and well written. However, English wording is partly not sufficient and some language editing will be required (for example in lines 15: ". . . interpretation this information . . .", ". . . highland over the world . . .", line 21 "its sounding regions."). There are three major deficiencies which need to be addressed before publication:

1) The implications of the Kesang Cave record for the reliability of the Guliya ice core chronology are barely touched in the manuscript. The issue is mentioned but not explained in detail. Unexperienced readers will not understand the point. So, why is Kesang Cave and also the new study supporting the opinion that the Guliya ice core chronology is not correct. What is the evidence from Kesang Cave? This is not explained in sufficient way.

[Figure]

2) The authors state that the Chongce ice cap is not older than 42 ka. They also argue that this age is much younger than those assumed for the lower parts of the Guliya ice cores. However, what are the paleoclimatic implications of their findings for the Chongce ice cap? Are the new data evidence for an ice-free region in the Chongce region in Marine Isotope Stage (MIS) 3? If so, what are the implications for the snow and ice accumulation rate at Chongce since the establishment of the ice cap sometime in MIS 3 or later? What does the statement that Chongce subglacial sediments are much younger than Guliya basal ice imply? Are the two ice caps comparable in terms of altitude, exposure, underlying relief, etc.?).

3) The authors state that the bottom sediments beneath Chongce ice cap are a combination of sediment and ice. What is the evidence that the base of the ice cap was actually reached? Are the sediments possibly representing a higher concentration of sediments within the ice but not necessarily basal sediments? The authors do not state that bedrock was drilled.

Minor comments:

Are lines 9-14 on page 2 relevant? They could be removed. Page 2, line 26: ice or water content? make clear Page 3, line 20: what is "obvious"? Page 3, line 20: what are the dots in the unit here? Page 4, line 9: What is the result if the first case is assumed? Explain the age result for this scenario too. Page 4, lines 11-12: how is the study of Takeuchi et al. related with the new study here? Page 6, line 17: abbreviation should be probably "Geochron." Page 7, line 7: no issue numbers Page 7, line 30: no capitalized letters if not for names or at beginning of sentence
* * *

---

## Referee Comment (RC2) · Anonymous Referee #2 · 21 Jun 2017

General comments:

The Manuscript by Zhang et al. is interesting, original and well written and suitable for publishing in the cryosphere after a few minor adjustments.

Specific comments:

1)The inconsistency in chronology between the Guliya ice core record and the Kesang stalagmite mentioned in the introduction, should be described.

2)It seems that the dating has been performed on basal ice, however it is a little unclear and should be more clear!

3)"Ice content" and "water content" seem to be used randomly. This should be more

clear.

4)The influence of the ice matrix on the dose-rate should be accounted for in detail and explained and an evaluation of dose-rate for each scenario should be performed.

5)The photograph of the Core 2 show a very clear transition to basal ice in the core, however Core 4+5 are retrieved at a different place at the ice-cap where the contour-lines in the map of Figure S1 suggest more ice dynamics, and the bottom part of the cores can be much more mixed. The 4-11 micro-metres fin-grained quartz used for the dating could be eolean material deposited onto the ice and therefore younger than the ice-cap. If this is the case, the grains would recieve most of their dose after mixing with the basal ice. The authors should discuss this possibillity.

6)In the conclusion the authors suggest collecting more suitable glacier basal sediment. It should be explained what "suitable" means.

Technical corrections:

page 1, line 9: more than one order of magnitude younger page 1, line 15: interpretation of this information. page 5, line 5-6: The sentence "We have no information about the behavior of ice in the sediment" should be refrased.

---

## Author Comment (AC1) · 3 Aug 2017

Dear J. Gombiner,

Many thanks for the constructive comments. Below I have made a point-to-point response to the comments. The comments are in black, and our response is in blue. I hope that the response can be acceptable.

Sincerely yours,

Hou Shugui
* * *
This is an interesting paper that should eventually be published.

However, the authors could give more thought to the calculation of dose rate and the meaning of the optical age for basal sediment.

The OSL age is the equivalent dose divided by the dose rate. In calculating the OSL age, the authors calculated a lower limit age for dehydrated sediment, containing air in the pore space, and an upper limit age, for hydrated sediment containing water in the pore space. The dose rate is lower for hydrated sediment because water attenuates radiation transfer from grain to grain. The actual sample came from sediment embedded within ice. The authors should calculate a dose rate for the real situation of sediment in ice.

There are potentially two additional sources of radiation that are not included in the dose rate calculation.

(1) Radiation from the bedrock or subglacial sediment.

(2) Radiation from dust layers in the core.

The authors should add these sources of radiation to the dose rate, or show that they are insignificant. If these other sources are included, the higher radiation dose rate would lower the calculated age.

Yes we fully agree that the dose rate is determined by many factors, including the potentially two additional sources of radiation as indicated above.

Willerslev et al. (2007) provided the first luminescence measurements on the single grains of quartz and feldspar extracted from a sample cut out of an opaque part of the Greenland Dye 3 basal ice containing dispersed sandy and silty particles. They found that the dose rate contribution from the underlying bedrock was negligible following calculations based on

radiation transport modelling software (MCNP5). Because no literature values are available to calculate the dose rate contribution from the underlying bedrock at our drilling site, we, for the moment, assume that its contribution to the dose rate was insignificant.

The sediment sample was collected from the very bottom several centimeters of Core 4. Its high particle content (~70%) suggests a similar condition as shown by the inset photo of Figure S1. Though dust layers are frequently observed along the Core 4, they are much weaker than the bottom section, as shown by the photo below with typical dust layers along the core.

[Figure]

A 18. 7 m ice core drilled at the summit (6530 m a.s.l.) of the Chongce ice cap in 1992 gives a maximum dust mass concentration of 955 mg kg-1 (Li et al., 2006). This provides a general impression of dust layers along the Chongce ice cores, which is ~3 orders of magnitude lower than the bottom sediment, suggesting insignificant influence of radiation from dust layers in the core, given a similar radiation intensity of the dust layers in the core and the bottom sediment.

As discussed above, slightly increased dose rate would be expected if these additional sources of radiation were included, thus resulting in a slightly younger age. Therefore, our upper limit age may be over estimated.

Li Y., Yang Y., Han J., Xie Z., M. Nakawo, K. Goto-Azuma. Persistent decrease of dust burden for about 100 years over middle-upper Troposphere of the southern Taklimakan Desert , China. J. Glaciol. Geocryol., 28, 873-878, 2006. (in Chinese with English abstract)

Willerslev, E., Cappellini, E., Boomsma, W., Nielsen, R., Hebsgaard, M. B., Brand, T. B., Hofreiter, M., Bunce, M., Poinar, H. N., Dahl-Jensen, D., Johnsen, S., Steffensen, J. P., Bennike, O., Schwenninger, J.-L., Nathan, R., Armitage, S., de Hoog, C.-J., Alfimov, V., Christl, M., Beer, J., Muscheler, R., Barker, J., Sharp, M., Penkman, K. E. H., Haile,

J., Taberlet, P., Gilbert, M. T. P., Casoli, A., Campani, E., and Collins, M. J.: Ancient biomolecules from deep ice cores reveal a forested southern Greenland, Science, 317, 111-114, doi: 10.1126/science.1141758, 2007.

Finally, I am not sure that the OSL age of the basal sediment directly relates to the age of the ice cap. The authors suggest that the sand-sized quartz grains are sourced from subglacial erosion. If true, it seems likely that some of the silt-sized quartz is also derived from subglacial erosion. Thus, it is conceivable that the dated aliquots are a mixture of eolian quartz and subglacially derived quartz.

Yes the dated aliquots are a mixture of eolian quartz and subglacially derived quartz. But because the distance from the summit of the Chonce ice cap to the drilling site is only several kilometers, and the ice cap is much shallower in comparison to the ice sheets, the scoured sediment experienced weak grinding. This suggests that even some of the silt-sized quartz is also derived from subglacial erosion, its portion might be very small. Thus the dated aliquots are mostly an eolian origin.

If the ice flow at the core site is dominated by downward vertical motion, then the OSL age of the eolian component of the dated aliquots would represent the time for the ice to move from the surface to the bed, not the age of the ice cap itself.

Yes we agree with the comment, and this will be clarified in the revision.

---

## Author Comment (AC3) · 3 Aug 2017

Dear Referee,

Many thanks for the constructive comments. Below I have made a point-to-point response to the comments. The comments are in black, and our response is in blue. I hope that the response can be acceptable.

Sincerely yours,

Hou Shugui
* * *
General comments:

The Manuscript by Zhang et al. is interesting, original and well written and suitable for publishing in the cryosphere after a few minor adjustments.

Specific comments:

1) The inconsistency in chronology between the Guliya ice core record and the Kesang stalagmite mentioned in the introduction, should be described.

We include a short introduction about the records of the Kesang Cave and the Guliya ice core in the Supplement in order to make the communication as concise as possible.

The Guliya ice core and the Kesang Cave core

In 1992, a 308.6 m ice core to bedrock was recovered from the Guliya ice cap located at 35°17'N, 81°29'E on the northwest Tibetan Plateau (Figure 1). The drilling site is at an elevation of 6200 m a.s.l. Top 266 m of the core was dated to a period spanning 110 ka, and ice below 290 m depth was suggested to be more than 500 ka old due to $^{36}$Cl-dead in the ice (Thompson et al., 1997). Three Guliya interstadials (Stages 3, 5a, and 5c) are marked by increases in $\delta^{18}$O values similar to that of the Holocene and Eemian (~124 ka ago) (Thompson et al., 1997).

The Kesang Cave is located in the Tekesi County, western China (42°52′ N, 81°45′ E, elevation ~2000 m a.s.l.) (Figure 1). Eight samples from the Kesang Cave were collected to establish the Kesang $\delta^{18}$O record with three covering the Holocene and five covering the rest of the Pleistocene portion. Cheng et al. (2012) obtained precise ages (~150 dates), all in stratigraphic order within errors, using a $^{230}$Th dating technique in the University of

Minnesota. The stalagmite $\delta^{18}O$ variations largely reflect changes in the $\delta^{18}O$ of meteoric precipitation (Cheng et al., 2012).

To reconcile the difference in the $\delta^{18}O$ variations between the Guliya and the Kesang records, Cheng et al. (2012) suggested that the Guliya record needs to be younger about a factor of two.

2) It seems that the dating has been performed on basal ice, however it is a little unclear and should be more clear!

Yes the dating was performed on the basal sediment. In fact, this sediment is a mixture of particles and ice. We will clarify this content in the revision.

3) "Ice content" and "water content" seem to be used randomly. This should be more clear.

We took a small portion of the sediment (13.4 g) for measuring ice content, which is determined by weighting the mass before and after drying, resulting in ~30% ice (water equivalent) content.

4) The influence of the ice matrix on the dose-rate should be accounted for in detail and explained and an evaluation of dose-rate for each scenario should be performed.

The infinite matrix dose rate was estimated using concentration-to-dose rate conversion constants presented by Adamiec and Aitken (1998) and the estimate of the dilution of the external dose rate by ice was assumed to be consistent with calculations recommended by Aitken (1985). Water, if present in the sediment matrix, absorbs radiation differently from mineral sediment, and has to be accounted for in the dose-rate calculations. Since we have, for the moment, no information about the influence of the ice matrix on the dose-rate, we use two extreme cases as our bounding scenarios, i.e., no water under the frozen condition and 30% water content if the sediment is saturated with water. The latter case (with high water content) results in a lower dose rate. Thus our upper limit age may be over estimated.

Adamiec, G. and Aitken, M.J.: Dose-rate conversion factors: update, Anc. TL, 16, 37-50, 1998.

Aitken, M. J.: Thermoluminescence dating, Academic Press, London. 1985.

5) The photograph of the Core 2 show a very clear transition to basal ice in the core, however Core 4+5 are retrieved at a different place at the ice-cap where the contourlines in the map of Figure S1 suggest more ice dynamics, and the bottom part of the cores can be much more mixed. The 4-11 micro-metres fin-grained quartz used for the dating could be eolean material deposited onto the ice and therefore younger than the ice-cap. If this is the case, the grains would recieve most of their dose after mixing with the basal ice. The authors should discuss this possibillity.

Yes the 4 -11 μm fine quartz grains used for the dating are mostly an eolian origin. The OSL age of the eolian component would represent the time for the ice to move from the surface to the bed, which is younger than the ice cap. In fact, this OSL age, as an upper limit, does not imply for an ice-free region in the Chongce region, but for an retreat of the ice cap above the elevation of the bottom at the drilling site during a (or more) warm period (or periods) since the upper limit age (e.g., MIS3, the Bølling-Allerød period, Holocene Climate Optimum). Because only limited results are gained, and many processes (each with its uncertainty) are involved in affecting the final age, we are cautious to avoid over-explaining the results at this moment.

6) In the conclusion the authors suggest collecting more suitable glacier basal sediment. It should be explained what "suitable" means.

We have drilled ice cores from several glaciers and ice caps on the Tibetan Plateau. This is the first time to collect sufficient amount of sediment at the Chongce ice core bottom for the luminescence dating. To avoid misunderstanding, we revised this sentence as the following.

The major limitation of the current work is the very small number of absolute datings. Future work should include collecting more glacier basal sediment samples for the luminescence dating....

Technical corrections:

page 1, line 9: more than one order of magnitude younger

Revised accordingly.

page 1, line 15: interpretation of this information.

Revised accordingly.

page 5, line 5-6: The sentence "We have no information about the behavior of ice in the sediment" should be refrased.

We revised this sentence as the following.

Since we have, for the moment, no information about the influence of the ice matrix on the dose-rate, we use two extreme cases as our bounding scenarios, i.e., no water under the frozen condition and 30% water content if the sediment is saturated with water.

---

## Editor Decision (ED1)

Dear authors,

Thanks for the detailed comments to the three reviews of your paper.

All three reviewers have provided constructive comments, as well as major questions and concerns with regards to your results and conclusions. Please find my summary of the most important points, as well as additional editorial comments below.

If you can provide sufficient clarification on these points, please provide a point to point reply, as well as a revised version of your paper (tracked change modus). Your reply and revised version will potentially be reviewed again.

Many thanks,

Julia

Text in *italics and blue* refers to the reply of the authors to the reviewers' comments or the original paper.

1. **Additional radiation sources, such as bedrock or dust layers**

The request by reviewer 1 was to give more thought to the calculation of dose rate and the meaning of the optical age for basal sediment including additional radiation sources from (1) bedrock or subglacial sediment and (2) from dust layers in the core.

Based on results from Greenland ice core (Willerslev et al., 2017) and dust layers from a different ice core (on the Chongce ice cap), you come to the conclusion that the dose rate contribution from the underlying bedrock and dust layers are negligible.

*Authors reply: Yes we fully agree that the dose rate is determined by many factors, including the potentially two additional sources of radiation as indicated above...... the moment, assume that its contribution to the dose rate was insignificant.*

Editor comment: Please also provide further information on your sampled core and the dose rate methods:

- If bedrock is incorporated in the basal glacier, these bedrock fractions might still influence the sample to some extend; please discuss this with respect to your glacier site and sampling depth.

- Information on dust layers from your core; if these are located close to sampled material

(in "normal" sediment 30 cm, in ice more), then this contributes to the dose rate and needs to be measured or considered.

- The process of embedding grains in ice is important. The radiation sources (radionuclides) may be separated from quartz/feldspar grains by ice increasing the distance between radiation source and dosimeter and, hence, reducing the dose rate by an extent that cannot be covered by measuring gamma-spec dry/unfrozen material.

- How are the high ice content/layers/lenses accounted for (varying densities)?

- It is not clear why "dehydrated" dose rates and respective ages are calculated considering that the material is an ice core (see also point 3).

- Please provide further clarification on the dose rate modeling using attenuation due to water and whether this can be transferred to the effect of ice, and how this dose rate measurement is representative of the natural condition within the ice core. Assuming water instead of ice for corrections of the dose rate efficiency may induce errors. The effective dose rate could also be (largely) overestimated and ages could be older (see also point 3).

**2. Sources of the dated material  (glacial erosion and eolian mixture) and relationship to age of ice cap**

This concern is raised by all reviewers and you do not provide any further details in your response.

Reviewer 1: The authors suggest that the sand-sized quartz grains are sourced from subglacial erosion. If true, it seems likely that some of the silt-sized quartz is also derived from subglacial erosion. Thus, it is conceivable that the dated aliquots are a mixture of eolian quartz and subglacially derived quartz.

*Author's reply: This suggests that even some of the silt-sized quartz is also derived from subglacial erosion, its portion might be very small. Thus the dated aliquots are mostly an eolian origin.*

Editor comment: This explanation is weak, because no big distance is needed to produce silt sized grains, it also could happen in just a freeze-thaw-cycle. It is thus important to investigate their equivalent dose distributions. If silt is only or predominantly from eolian transport, one distinct, normally distributed population can be expected. Grains from grinding material below the glacier was not bleached before getting into the basal glacier part and hence would form an different population.

Please provide further clarification.

Reviewer 1: If the ice flow at the core site is dominated by downward vertical motion, then the OSL age of the eolian component of the dated aliquots would represent the time for the ice to move from the surface to the bed, not the age of the ice cap itself.

*Author's reply: Yes we agree with the comment, and this will be clarified in the revision.*

Editor comment: The ice cap may have existed at this place over a longer period of time and OSL ages give only a sort of transition/travel time of ice portions/sediment particles from top/incorporation to the bottom layer. As a result, a minimum age estimate can be given, but the age of the ice cap could be much older. Please provide further details.

Reviewer 1 and 3: If the ice flow at the core site is dominated by downward vertical motion, then the OSL age of the eolian component of the dated aliquots would represent the time for the ice to move from the surface to the bed, not the age of the ice cap itself.

Editor comment: In your reply to reviewer 3 you confirm that that the 4 -11 µm fine quartz grains used for the dating are mostly of eolian origin. The OSL age of the eolian component would represent the time for the ice to move from the surface to the bed, which is younger than the ice cap.

*Author's reply: The OSL dating results of the coarse grain (90-150µm) quartz are shown below. Water content is assigned an absolute uncertainty of ±7%. The slightly older ages of the coarse grains in comparison to the fine grain quartz may imply that the former were more affected by the local scoured particles that were partly bleached. Another disadvantage for the coarse grain aliquots is that their medium and slow components accounts for a significant part of the natural OSL signal.*

| Sample | U (ppm) | | Th (ppm) | | K (%) | | Water content | D$_e$ | Dose rate | Age |
|---|---|---|---|---|---|---|---|---|---|---|
| | Gamma[1] | NAA | Gamma | NAA | Gamma | NAA | (%) | (Gy) | (Gy/ka) | (ka) |
| CCICE | 3.66±0.15 | 3.45±0.12 | 11.21±0.42 | 11.40±0.32 | 3.52±0.10 | 3.48±0.08 | 0 | 238±51 | 5.25±0.45 | 45 ±11 |
| CCICE | 3.66±0.15 | 3.45±0.12 | 11.21±0.42 | 11.40±0.32 | 3.52±0.10 | 3.48±0.08 | 30 | 238±51 | 3.85±0.24 | 62 ±14 |

Editor comment: (1) Are those assumed to be eolian, too? Or what is the assumed incorporation process and, hence, bleaching? After reading the above comments, I would have expected that only the fine grained fraction can represent eolian sediments and, hence, no basal material, and consequently the coarse grained fraction should be much older. If the coarse grains have different properties (i.e. medium and slow components) then they are likely derived from a different source. (2) Table above: how can the water content be 0% in this type of sediment?

*Author's reply: Yes the 4 -11 μm fine quartz grains used for the dating are mostly an eolian origin. The OSL age of the eolian component would represent the time for the ice to move from the surface to the bed, which is younger than the ice cap. In fact, this OSL age, as an upper limit, does not imply for an ice-free region in the Chongce region, but for an retreat of the ice cap above the elevation of the bottom at the drilling site during a (or more) warm period (or periods) since the upper limit age (e.g., MIS3, the Bølling-Allerød period, Holocene Climate Optimum). Because only limited results are gained, and many processes (each with its uncertainty) are involved in affecting the final age, we are cautious to avoid over-explaining the results at this moment.*

Editor comment: This is contractionary, because above you say the OSL ages refer to eolian grains travelling from the ice top to the bottom, which happens without ice retreat?

**3. Water/ice content (issues raised by reviewers 1 and 3)**

Reviewer 3: The actual sample came from sediment embedded within ice. The authors should calculate a dose rate for the real situation of sediment in ice.

Editor comment: This comment has not been addressed in the reply. The authors calculate two extreme ranges (dry to 30% water content). Why would the dehydrated scenario be realistic for any time slice of the sediment's age? How was the 30% water content chosen?

You should explain under which conditions the sediment was dehydrated and how likely this conditions persisted - as long as the cold ice/glacier is on top the sediment is frozen or retrieves melt-water from the glacier.

*Author's reply: no water under the frozen condition and 30% water content if the sediment is saturated with water.*

Editor comment: Please be specific and correct in your wording. Does frozen mean subzero temperatures with no liquid water or ice content? Please differentiate between the phases water and ice and consider that water also exists in ice-sedimentary material at zero and sub zero temperatures.

*Author's reply: The latter case (with high water content) results in a lower dose rate. Thus our upper limit age may be over estimated.*

Editors comment: Why? Because the 30% water content are unrealistic?

4. **Further editorial comments**

- I would prefer "maximum or minimum ages" instead of "upper limit".
- Reference to Adamiec and Aitken (1998): This is a standard procedure and not specifically addressing samples in/under ice.

- Drilling of the 2012 ice cores is described in the introduction, as well as the two 2013 cores. It should be clearly described which data are presented from which core and which cores are used for illustrative purposes. For example, an image of the basal core material is shown for core location 2, but dating has been carried out with cores 3 and 4. Was dating done on the basal material of both cores?
  This detailed information needs to be included in the method section.

- *"Electron microscope of the sediment of coarse grain quartz might have scoured from bedrock, but the finer grained quartz was chosen as appropriate for luminescence dating of our sample (hence excluded the coarse grain quartz from further calculation). In the end, 8 fine grain quartz aliquots were used for OSL age calculation, resulting in an average equivalent dose (De) of 178 $\pm$ 9 Gy (Table 1)."*

Why not use the coarse grain quartz if this is the material clearly defined as bedrock? Please clarify.

Please also do not use phrases such as ", *we believe that only the fine grained…*", instead use data to support your statement.

- *"Only the aliquots with recycling ratios within the acceptable range were used for further analysis. Among these, one additional aliquot was also excluded because its De value fell outside 2$\sigma$ of the distribution (Fig. 3)."*

Can you give further information about this sample? Is it usual to remove a sample if the value falls outside 2$\sigma$ of the distribution (Fig. 3)?

- Table 1 provides results from sample CCICE – please clarify the abbreviation. Where do you provide the information on the 8 aliquots?
- Figures, Tables and legends (in Paper and Supplement)
  All figures and tables need to include the information for understanding the figure. Please provide detailed figure and table legends. For example, Figure S1 shows 5 coring locations. When were they drilled? What results are used from which core in the paper? The paper mentions 4 ice cores, what about core 5?
  The data shown are from which ice core specifically?

---

## Author Response (AR2)

Dear Editor,

Many thanks for your further constructive comments. Below I have made a point-to-point response to the comments. The comments are in black, and our response is in blue. I have revised the original manuscript following the comments.

As pointed out in the paper, there is only one study on the luminescence dating of a Greenland ice core so far. Here we present the first luminescence dating of Tibetan glacier basal sediment, thanks to the rare chance to collect sufficient basal sediment for such kind of work. We agree that the current work is quite preliminary, and there exists much uncertainty, such as the dose rate, additional radiation sources from bedrock and dust layers in the core, specific processes underneath the glacier, and so on. For this reason, the dose rate we adopted to calculate the age might be smaller than the real one. So we only provide an upper constraint for the age of the sediment.

We have recently got two pieces of independent evidence that back up our conclusions in this manuscript. The figure below shows the AMS $^{14}$C age of 22 and 9 samples along the Chongce 216.6 m and the 135.8 m ice cores, respectively, with the oldest AMS $^{14}$C age of ~ 6.25 ka.

[Figure]

We have also measured the $\delta^{18}O$ values of the upper and bottom sections of the Chongce 135.8 m core. It's apparent that no sharp decrease of $\delta^{18}O$ can be observed for the bottom section, suggesting that no glacial ice preserves in the bottom ice.

[Figure]

CB-6: 3112±186 cal yr BP
CB-8: 4705±251 cal yr BP
CB-9: 6253±226 cal yr BP

Unfortunately, I can not include these dataset in the manuscript because they have not been published yet. With the OSL dataset only, we are able to give an upper age constraint of 42 ± 4 ka, which is still more than one order of magnitude younger than

the previously suggested age of the glacier basal ice from the same area. Our work might bring forward potential method for such a dating purpose.

Below I have made a point-to-point response to the comments by the reviewers, and the Editor comments. The comments are in black, and our response is in blue. I have accordingly revised the original manuscript following these comments.

Sincerely yours,

Hou Shugui
* * *
1. Additional radiation sources, such as bedrock or dust layers

Editor comments:

Please also provide further information on your sampled core and the dose rate methods:

- If bedrock is incorporated in the basal glacier, these bedrock fractions might still influence the sample to some extend; please discuss this with respect to your glacier site and sampling depth.

Yes we agree that these bedrock fractions might still influence the sample to some extend. From Fig. S2, we notice that fast component of the fine grain aliquots accounts for around 87% of the natural OSL signal, indicating that the signal is fast component dominant. If for a high bedrock fraction, medium and slow components should be high due to no or less bleaching of the bedrock, similar to what was

observed for the OSL decay curves of the coarse grain aliquots, displaying a relatively slow trend (Fig. 2).

[Figure]

Figure S2: Natural OSL decay curves and their relative components for the fine grain quartz aliquots. Sum, F, M and S represent natural OSL signal, fast, medium and slow components, respectively. Fitting curve is matched with "Luminescence Analyst" program.

[Figure]

Figure 2: The OSL dose-response and decay curves of the coarse and fine grain quartz aliquots. Inserts show decay curves of the natural and regenerated dose (2000β) of the sample.

- Information on dust layers from your core; if these are located close to sampled material (in "normal" sediment 30 cm, in ice more), then this contributes to the dose rate and needs to be measured or considered.

This "subglacial sediment" is what we measured for luminescence dating.

- The process of embedding grains in ice is important. The radiation sources (radionuclides) may be separated from quartz/feldspar grains by ice increasing the distance between radiation source and dosimeter and, hence, reducing the dose rate by an extent that cannot be covered by measuring gamma-spec dry/unfrozen material.

Yes, but we do not have evidence about the process at this moment. For this reason, the dose rate we adopted to calculate the age might be smaller than the real one. So we only provide an upper constraint for the age of the sediment, which is still more than one order of magnitude younger than the previously suggested age of the glacier basal ice from the same area. As mentioned above, we have recently got independent evidence from AMS [14]C age and the stable isotopic profile of the Chongce ice cores, which back up our conclusions but also give a more precise constraint.

- How are the high ice content/layers/lenses accounted for (varying densities)?

It's well recognized that water content is one of the major uncertainty factors of the Luminescence dose rate calculation because we do not know the real situation of sediments during geological history. To approach this, two extreme situations for the sediment were presumed, one is zero water content which represents no ice (or water), and the other is the sediment full immersed by the water (saturation water content), i.e., $30 \pm 15\%$ water content in our case. Under these two situations, the maximum and minimum dose rates from water content were estimated.

- It is not clear why "dehydrated" dose rates and respective ages are calculated considering that the material is an ice core (see also point 3).

The sediment that we measured for luminescence dating is from the very bottom of the Chongce 216.6 m ice core. It's in fact a mixture of ~30% ice (water equivalent)

and consequently ~70% sediment. To simplify the calculation of dose rates, we consider two extreme situations as mentioned above.

- Please provide further clarification on the dose rate modeling using attenuation due to water and whether this can be transferred to the effect of ice, and how this dose rate measurement is representative of the natural condition within the ice core. Assuming water instead of ice for corrections of the dose rate efficiency may induce errors. The effective dose rate could also be (largely) overestimated and ages could be older (see also point 3).

We totally agree with these comments. By comparison of the luminescence age and our recently measured [14]C results, the effective dose rate was overestimated and ages were older. But we do not have evidence to decipher this effect, so we only provide an upper constraint for the age of the sediment based on the luminescence measurements only.

Finally, I am not sure that the OSL age of the basal sediment directly relates to the age of the ice cap. The authors suggest that the sand-sized quartz grains are sourced from subglacial erosion. If true, it seems likely that some of the silt-sized quartz is also derived from subglacial erosion. Thus, it is conceivable that the dated aliquots are a mixture of eolian quartz and subglacially derived quartz.

Yes the dated aliquots are a mixture of eolian quartz and subglacially derived quartz. From Fig. S2, we notice that fast component of the fine grain aliquots accounts for around 87% of the natural OSL signal, indicating that the signal is fast component dominant. If for a high bedrock fraction, medium and slow components should be high due to no or less bleaching of the bedrock, similar to what was observed for the OSL decay curves of the coarse grain aliquots, display ing a relatively slow trend (Fig.

2). This suggests that even some of the silt-sized quartz can be derived from subglacial erosion, its portion might be very small. Thus the dated aliquots are mostly an eolian origin.

**2. Sources of the dated material (glacial erosion and eolian mixture) and relationship to age of ice cap**

Editor comment: This explanation is weak, because no big distance is needed to produce silt sized grains, it also could happen in just a freeze-thaw-cycle. It is thus important to investigate their equivalent dose distributions. If silt is only or predominantly from eolian transport, one distinct, normally distributed population can be expected. Grains from grinding material below the glacier was not bleached before getting into the basal glacier part and hence would form an different population.

Sharp and Gomez (1986) suggested that quartz particles should be "tools" producing comminution of softer minerals within subglacial environments while they themselves remain unaltered. We have noticed the work by Wright (1995), who performed glacial grinding simulations by using a ring shear box designed to examine abrasion and size reduction of quartz grains under "grinding" conditions analogous to those experienced in a subglacial environment. The findings from experimental runs suggest that, although particle breakage and comminution does occur, little silt-sized material is actually produced. The amount of silt produced ranges from approximately 9% of the original sample in one experimental run to less than 1% for the other seven runs. Our direct evidence from Fig. S2, we notice that fast component of the fine grain aliquots accounts for around 87% of the natural OSL signal, indicating that the signal is fast component dominant. If for a high bedrock fraction, medium and slow components should be high due to no or less bleaching of the bedrock. So the fine quartz grains used for the dating are mostly an eolian origin.

Sharp, M. and Gomez, B.: Processes of debris comminution in the glacial environment and implications for quartz sand-grain micromorphology. Sedimentary Geology, 46, 33-47, 1986.

Wright, J. S.: Glacial comminution of quartz sand grains and the production of loessic silt: a simulation study, Quat. Sci. Rev., 14, 669-680, 1995.

Editor comment: The ice cap may have existed at this place over a longer period of time and OSL ages give only a sort of transition/travel time of ice portions/sediment particles from top/incorporation to the bottom layer. As a result, a minimum age estimate can be given, but the age of the ice cap could be much older. Please provide further details.

Reviewer 1 and 3: If the ice flow at the core site is dominated by downward vertical motion, then the OSL age of the eolian component of the dated aliquots would represent the time for the ice to move from the surface to the bed, not the age of the ice cap itself.

Editor comment: In your reply to reviewer 3 you confirm that the 4 - 11 μm fine quartz grains used for the dating are mostly of eolian origin. The OSL age of the eolian component would represent the time for the ice to move from the surface to the bed, which is younger than the ice cap.

Yes the fine quartz grains used for the dating are mostly an eolian origin, implying that the luminescence dating would represent the time for the ice to move from the surface to the bed. Our new data does not imply for an ice-free region in the Chongce region since 42 ± 4 ka ago, but for an ice-free condition below the elevation of the bottom at the drilling site during a (or more) warm period (or periods) since this upper

constraint age (e.g., Marine Isotope Stage (MIS) 3, the Bølling-Allerød period, Holocene Climate Optimum). Takeuchi et al. (2014) reported radiocarbon dating of organic soil from the bottom of an 86.87 m ice core drilled at the top of the Grigoriev Ice Cap (41°58'33" N, 77°54'48" E) in the Tien Shan Mountains, showing that the age of the soil is 12 656 - 12 434 cal years before present. Therefore, Takeuchi et al. (2014) suggested that the Grigoriev Ice Cap did not exist in the Bølling-Allerød period, which gives support to our results.

Editor comment: (1) Are those assumed to be eolian, too? Or what is the assumed incorporation process and, hence, bleaching? After reading the above comments, I would have expected that only the fine grained fraction can represent eolian sediments and, hence, no basal material, and consequently the coarse grained fraction should be much older. If the coarse grains have different properties (i.e. medium and slow components) then they are likely derived from a different source. (2) Table above: how can the water content be 0% in this type of sediment?

We calculated the OSL dating results of the coarse grain (90 - 150μm) quartz upon requested by the reviewer, for comparison purpose. We do not think that the coarse grains are suitable for OSL dating due to high medium and slow components. For this reason, we gave up calculating the OSL dating results of the coarse grains in the manuscript. So we also gave up further consideration of processes concerning to the coarse grains.

Editor comment: This is contractionary, because above you say the OSL ages refer to eolian grains travelling from the ice top to the bottom, which happens without ice retreat?

During the warm period (or periods) since our upper constraint age (e.g., Marine Isotope Stage (MIS) 3, the Bølling-Allerød period, Holocene Climate Optimum), glaciers retreated to a minimum area. Afterwards, with decreasing temperature and/or increasing precipitation, glaciers expanded, so the OSL ages refer to eolian grains travelling from the ice top to the bottom during the glacier expansion. As we discussed before, there are other radiation resources that we did not include and uncertainties for calculating the dose rate, so the OSL age can not be regarded straightforwardly as the time of glacier expansion.

3. Water/ice content (issues raised by reviewers 1 and 3)

Reviewer 3: The actual sample came from sediment embedded within ice. The authors should calculate a dose rate for the real situation of sediment in ice.

Editor comment: This comment has not been addressed in the reply. The authors calculate two extreme ranges (dry to 30% water content). Why would the dehydrated scenario be realistic for any time slice of the sediment's age? How was the 30% water content chosen?
You should explain under which conditions the sediment was dehydrated and how likely this conditions persisted - as long as the cold ice/glacier is on top the sediment is frozen or retrieves melt-water from the glacier.

We first took a small portion of the sediment sample for measuring ice content, which is determined by weighting the mass before and after drying, resulting in ~30% ice (water equivalent) content. As the water content may have varied over the entire burial period of the sample, we assigned a relative uncertainty of 50% to allow for possible fluctuations.

This scenario may apply during the warm period (or periods) since our upper constraint age (e.g., the Bølling-Allerød period, Holocene Climate Optimum). There are more and more evidence that glaciers on the Tibetan Plateau, Tienshan, Altai might retreat seriously or even disappeared during the Bølling-Allerød period, Holocene Climate Optimum. For instance, Takeuchi et al. (2014) suggested that the Grigoriev Ice Cap did not exist in the Bølling-Allerød period. Herren et al. (2013) suggested the onset of Neoglaciation 6000 years ago in western Mongolia Altai. Thompson et al. (2005) suggested the Dasuopu ice field in the Himalayas and the Puruogangri ice field in the central Tibetan Plateau accumulated during the warm (moist) Holocene period and is thus a neoglacial feature.

Herren, P. A., Eichler, A., Machguth, H., Papina, T., Tobler, L., Zapf, A., and Schwikowski, M.: The onset of Neoglaciation 6000 years ago in western Mongolia revealed by an ice core from the Tsambagarav mountain range, Quaternary Sci. Rev., 69, 59–68, doi:10.1016/j.quascirev.2013.02.025, 2013.

Takeuchi, N., Fujita, K., Aizen, V. B., Narama, C., Yokoyama, Y., Okamoto, S., Naoki, K., and Kubota, J.: The disappearance of glaciers in the Tien Shan Mountains in Central Asia at the end of Pleistocene, Quat. Sci. Rev., 103, 26-33, doi: 10.1016/j.quascirev.2014.09.006, 2014.

Thompson, L. G., Davis, M. E., Mosley-Thompson, E., Lin, P., Henderson, K. A., and Mashiotta, T. A., Tropical ice core records: evidence for asynchronous glaciation on Milankovitch timescales, J. Quaternary. Sci., 20, 723-733, 2005.

Editors comment: Why? Because the 30% water content are unrealistic?

The measured ice (water equivalent) content is ~30% in the sediment sample. This

does not mean that there is ~30% water underneath the glacier at present (in fact, it's frozen), but during the warm period(s) in history, there might be ~30% water at the bottom of the ice core drilling site. And because the water content might have varied over the entire burial period of the sample, we assigned a relative uncertainty of 50% to allow for possible fluctuations.

4. Further editorial comments

- I would prefer "maximum or minimum ages" instead of "upper limit".

It seems that there is a little difference between "upper limit" and "maximum" here. The real age is always younger than the "upper limit" (though we do not know exactly how much younger), but there is possibility that age can reach the "maximum" age.

To avoid misunderstanding, we changed "upper limit" to "upper constraint" in the revision.

- Reference to Adamiec and Aitken (1998): This is a standard procedure and not specifically addressing samples in/under ice.

This is shortcoming of the current work. We have no evidence about the specific procedure at our study site, so we considered two extreme situations to envelope all the possibilities during history.

- Drilling of the 2012 ice cores is described in the introduction, as well as the two 2013 cores. It should be clearly described which data are presented from which core and which cores are used for illustrative purposes. For example, an image of the basal core material is shown for core location 2, but dating has been carried out with cores 3 and 4. Was dating done on the basal material of both cores?

This detailed information needs to be included in the method section.

This detailed information is provided in caption of Fig. S1.

Figure S1: Map showing part of the Chongce ice cap where our ice cores were recovered. Core 1 (133.8 m) and Core 2 (135.8 m) to bedrock and Core 3 (58.8 m) were drilled at an altitude of 6010 m a.s.l. in 2012. Core 4 (216.6 m) and Core 5 (208.6 m) to bedrock were drilled at an altitude of 6100 m a.s.l. in 2013. The sediment samples that we measured for luminescence dating was collected from the very bottom section of Core 4, which is similar to the bottom several centimeters section as shown by the inset photo of Core 2.

Why not use the coarse grain quartz if this is the material clearly defined as bedrock? Please clarify.

The coarse grains may be more affected by the local scoured particles that were partly bleached. Another disadvantage for the coarse grain aliquots is that their medium and slow components accounts for a significant part of the natural OSL signal.

Please also do not use phrases such as"we believe that only the fine grained…", instead use data to support your statement.

We have revised this sentence, and other similar sentences as well.

These results imply that only the fine grain quartz is appropriate for luminescence dating....

Can you give further information about this sample? Is it usual to remove a sample if the value falls outside 2σ of the distribution (Fig. 3)?

We have given all the information about this sample that we have.

Yes it is conventional way to remove a sample if the value falls outside 2σ of the

distribution.

- Table 1 provides results from sample CCICE -please clarify the abbreviation. Where do you provide the information on the 8 aliquots?

CCICE stands for Chongce Ice.

Results on the 8 aliquots will be put into public data base.

- Figures, Tables and legends (in Paper and Supplement)

Yes details were included in the caption.

Figure S1: Map showing part of the Chongce ice cap where our ice cores were recovered. Core 1 (133.8 m) and Core 2 (135.8 m) to bedrock and Core 3 (58.8 m) were drilled at an altitude of 6010 m a.s.l. in 2012. Core 4 (216.6 m) and Core 5 (208.6 m) to bedrock were drilled at an altitude of 6100 m a.s.l. in 2013. The sediment samples that we measured for luminescence dating was collected from the very bottom section of Core 4, which is similar to the bottom several centimeters section as shown by the inset photo of Core 2.

Dear Referee,

Many thanks for the constructive comments. Below I have made a point-to-point response to the comments. The comments are in black, and our response is in blue.

Sincerely yours,

Hou Shugui
* * *
comments by Anonymous Referee #1

The manuscript by Zhang et al. provides interesting and new data which justify publication in The Cryosphere. It is relatively well structured and well written. However, English wording is partly not sufficient and some language editing will be required (for example in lines 15: "....    interpretation this information....", "....highland over the world....", line 21 "its sounding regions.").

We have accordingly revised these sentences.

Lines 14-15: We revised this sentence as:

A precise chronology is the essential first step for a reliable interpretation of the ice core records.

Lines 15-16: We revised this sentence as:

The Tibetan Plateau, sometimes called "the Roof of the World", is the world's highest and largest plateau with an average elevation exceeding 4000 m above sea level (a.s.l.) and an area of 2.5 Million square kilometers.

Lines 20-21: Following Cheng et al. GRL 2012, we revised this sentence as:

During the past two decades, the Guliya ice core record has been widely used as a benchmark for correlating regional climate variables in the Westerlies region of the central Asia and the northern Tibetan Plateau.

There are three major deficiencies which need to be addressed before publication:

1) The implications of the Kesang Cave record for the reliability of the Guliya ice core chronology are barely touched in the manuscript. The issue is mentioned but not explained in detail. Unexperienced readers will not understand the point. So, why is Kesang Cave and also the new study supporting the opinion that the Guliya ice core chronology is not correct. What is the evidence from Kesang Cave? This is not explained in sufficient way.

Yes we include a short introduction about the records of the Kesang Cave and the Guliya ice core in the Supplement in order to make the communication as concise as possible.

The Guliya ice core and the Kesang Cave core

In 1992, a 308.6 m ice core to bedrock was recovered from the Guliya ice cap located at 35°17'N, 81°29'E on the northwest Tibetan Plateau (Figure 1). The drilling site is at an elevation of 6200 m a.s.l. Top 266 m of the core was dated to a period spanning 110 ka, and ice below 290 m depth was suggested to be more than 500 ka old due to $^{36}$Cl-dead in the ice (Thompson et al., 1997). Three Guliya interstadials (Stages 3, 5a, and 5c) are marked by increases in $\delta^{18}$O values similar to that of the Holocene and Eemian (~124 ka ago) (Thompson et al., 1997).

The Kesang Cave is located in the Tekesi County,western China (42°52′ N, 81°45′ E, elevation ~2000 m a.s.l.) (Figure 1). Eight samples from the Kesang Cave were

collected to establish the Kesang $\delta^{18}$O record with three covering the Holocene and five covering the rest of the Pleistocene portion. Cheng et al. (2012) obtained precise ages (~150 dates), all in stratigraphic order within errors, using a $^{230}$Th dating technique in the University of Minnesota. The stalagmite $\delta^{18}$O variations largely reflect changes in the $\delta^{18}$O of meteoric precipitation (Cheng et al., 2012).

To reconcile the difference in the $\delta^{18}$O variations between the Guliya and the Kesang records, Cheng et al. (2012) suggested that the Guliya record needs to be younger about a factor of two.

2) The authors state that the Chongce ice cap is not older than 42 ka. They also argue that this age is much younger than those assumed for the lower parts of the Guliya ice cores. However, what are the paleoclimatic implications of their findings for the Chongce ice cap? Are the new data evidence for an ice-free region in the Chongce region in Marine Isotope Stage (MIS) 3? If so, what are the implications for the snow and ice accumulation rate at Chongce since the establishment of the ice cap sometime in MIS 3 or later? What does the statement that Chongce subglacial sediments are much younger than Guliya basal ice imply? Are the two ice caps comparable in terms of altitude, exposure, underlying relief, etc.?).

We have thought seriously about these constructive comments.

The luminescence dating of the basal sediment of the Chongce ice cap provides an upper constraint of $42 \pm 4$ ka. This might imply that the ice age at the bottom of the drilling site should be younger than this upper limit, although we do not know the exact age of the bottom ice at the drilling site. The new data does not imply for an ice-free region in the Chongce region in Marine Isotope Stage (MIS) 3, but for an ice-free condition at and below the elevation of the bottom at the drilling site during a (or more) warm period (or periods) since the upper limit age (e.g., MIS3, the

Bølling-Allerød period, Holocene Climate Optimum). A similar work was also performed for the disappearance of glaciers in the Tien Shan Mountains during the Bølling-Allerød period (Takeuchi et al., 2014).

Given the surface elevation of the Chongce drilling site of 6100 m a.s.l. and the ice core length of 216.6 m, the elevation at the bottom of the Chongce drilling site should be 5883.4 m a.s.l. As to the Guliya ice core, the surface elevation of the drilling site of 6200 m a.s.l. and the ice core length of 308.6 m result in an elevation at the bottom of the Guliya drilling site to be 5891.4 m a.s.l., suggesting that the age of the bottom ice at the Chongce and the Guliya drilling sites might be comparable. Thus our new data can not support the previously suggested age of more than 500 ka old at the Guliya ice core bottom (Thompson et al., 1997).

It's quite unlikely for serious glacier retreat during Marine Isotope Stage (MIS) 3, but there are independent evidence for disappearance of glaciers during the Bølling-Allerød period and Holocene Climate Optimum. To avoid misunderstanding, we deleted Marine Isotope Stage (MIS) 3 in the revision.

Takeuchi, N., Fujita, K., Aizen, V. B., Narama, C., Yokoyama, Y., Okamoto, S., Naoki, K., and Kubota, J.: The disappearance of glaciers in the Tien Shan Mountains in Central Asia at the end of Pleistocene, Quat. Sci. Rev., 103, 26-33, doi: 10.1016/j.quascirev.2014.09.006, 2014.

3) The authors state that the bottom sediments beneath Chongce ice cap are a combination of sediment and ice. What is the evidence that the base of the ice cap was actually reached? Are the sediments possibly representing a higher concentration of sediments within the ice but not necessarily basal sediments? The authors do not state that bedrock was drilled.

Yes bedrock was not drilled because the ice core drilling gear that we used is not able to drill through rock. As far as we know, the GISP2 ice core from the Greenland summit may be the only one with 2 m bedrock recovered. However, we have several pieces of evidences that the base of the ice cap might be actually reached.

① The bottom sediment is consisted of particles with wide range of size, including a high fraction of coarse particles. We had roughly measured the size distribution of a bottom sediment sample, with the results as shown in the table below. These coarse particles can not be an eolian origin. Moreover, the mountains surrounding the ice core drilling sites are snow and ice covered, thus eliminating the possibility that these coarse particles are from the surrounding high mountains. Therefore, these coarse particles should be scoured from the bed ground beneath the glacier, implying that the base of the ice cap might be actually reached.

| size (μm) | < 150 | 150 - 900 | 900 - 2000 | > 2000 |
|---|---|---|---|---|
| quality (g) | 4.0 | 4.28 | 3.97 | 9.68 |
| percent (%) | 18.2 | 19.5 | 18.1 | 44.1 |

② Due to the limit of luminescence test, we can not take a photo the bottom section of the Core 4, but the high particle content (~70%) of the bottom sediment of Core 4 suggests a similar condition as shown by the inset photo of Figure S1. Though dust layers are frequently observed along the Core 4, they are much weaker than the bottom section. The photo below with typical dust layers along the core, when compared to the inset photo of Figure S1, makes clear the uniqueness of the bottom sediment section.

[Figure]

A 18. 7 m ice core drilled at the summit (6530 m a.s.l.) of the Chongce ice cap in 1992 gives a maximum dust mass concentration of 955 mg kg$^{-1}$ (Li et al., 2006). This provides a general impression of dust layers along the Chongce ice cores, which is ~3 orders of magnitude lower than the bottom sediment, confirming the uniqueness of the bottom sediment section.

Li Y., Yang Y., Han J., Xie Z., M. Nakawo, K. Goto-Azuma. Persistent decrease of dust burden for about 100 years over middle-upper Troposphere of the southern Taklimakan Desert , China. J. Glaciol. Geocryol., 28(6), 873-878, 2006. (in Chinese with English abstract)

③ Result from ground penetrating radar gives an ice thickness of ~ 214 m at the drilling site, which is very close to our ice core length of 216.6 m.

④ We drilled the ice cores by using an electromechanical drill in a dry hole. Unfortunately, this kind electromechanical drill is not designed to penetrate into bedrock. But when the cutters of the drill reached the bedrock, a unique vibrating movement can be felt through the cable by our experienced engineer, who has experience of drilling ice cores for >20 years. Below I attach a photo of the cutters after the last run of drilling, showing the blade fractures that were caused by the high-speed spinning cutters bounced from the bedrock.

[Figure]

Minor comments:

Are lines 9-14 on page 2 relevant? They could be removed.

We revised this paragraph, and included this basic information in the caption of Fig. S1.

Page 2, line 26: ice or water content? make clear

The total bottom sediment including ice is 1431.7 g. We first took a small portion of the sediment (13.4 g) for measuring ice content, which is determined by weighting the mass before and after drying, resulting in ~30% ice (water equivalent) content.

Page 3, line 20: what is "obvious"?

We have revised this sentence to make it clear.

Samples with IRSL (infrared stimulated luminescence) vs. blue OSL signal ratios over 10% of unity would be re-treated with fluosilicic acid again until the ratio was within 10% of unity.

Page 3, line 20: what are the dots in the unit here?

We have revised it as "IRSL (infrared stimulated luminescence) vs. blue OSL signal ratios".

Page 4, line 9: What is the result if the first case is assumed? Explain the age result for this scenario too.

The OSL dating results of the coarse grain (90-150μm) quartz are shown below. Water content is assigned an absolute uncertainty of ±7%. The slightly older ages of the coarse grains in comparison to the fine grain quartz may imply that the former were more affected by the local scoured particles that were partly bleached. Another disadvantage for the coarse grain aliquots is that their medium and slow components accounts for a significant part of the natural OSL signal.

| Sample | U (ppm) | | Th (ppm) | | K (%) | | Water content | $D_e$ | Dose rate | Age |
|--------|---------|-----|----------|-----|-------|-----|---------------|-------|-----------|-----|
| | Gamma[1] | NAA | Gamma | NAA | Gamma | NAA | (%) | (Gy) | (Gy/ka) | (ka) |
| CCICE | 3.66±0.15 | 3.45±0.12 | 11.21±0.42 | 11.40±0.32 | 3.52±0.10 | 3.48±0.08 | 0 | 238±51 | 5.25±0.45 | 45 ±11 |
| CCICE | 3.66±0.15 | 3.45±0.12 | 11.21±0.42 | 11.40±0.32 | 3.52±0.10 | 3.48±0.08 | 30 | 238±51 | 3.85±0.24 | 62 ±14 |

Page 4, lines 11-12: how is the study of Takeuchi et al. related with the new study here?

Takeuchi et al. (2014) reported radiocarbon dating of organic soil from the bottom of an 86.87 m ice core drilled at the top of the Grigoriev Ice Cap (41°58'33" N, 77°54'48" E. Fig. 1) in the Tien Shan Mountains, showing that the age of the soil is 12 656 -12 434 cal years before present. Therefore, Takeuchi et al. (2014) suggested the disappearance of glaciers in the Tien Shan Mountains during the Bølling-Allerød period, which gives support to what we said in the manuscript.

"Our new data does not imply for an ice-free region in the Chongce region since 42 ± 4 ka ago, but for an ice-free condition below the elevation of the bottom at the drilling site during a (or more) warm period (or periods) since this upper constraint age (e.g., the Bølling-Allerød period, Holocene Climate Optimum)."

Page 6, line 17: abbreviation should be probably "Geochron."

Revised accordingly.

Page 7, line 7: no issue numbers

Revised accordingly.

Page 7, line 30: no capitalized letters if not for names or at beginning of sentence

Revised accordingly.

Dear Referee,

Many thanks for the constructive comments. Below I have made a point-to-point response to the comments. The comments are in black, and our response is in blue.

Sincerely yours,

Hou Shugui
* * *
General comments:

The Manuscript by Zhang et al. is interesting, original and well written and suitable for publishing in the cryosphere after a few minor adjustments.

Specific comments:

1)  The inconsistency in chronology between the Guliya ice core record and the Kesang stalagmite mentioned in the introduction, should be described.

We include a short introduction about the records of the Kesang Cave and the Guliya ice core in the Supplement in order to make the communication as concise as possible.

The Guliya ice core and the Kesang Cave core

In 1992, a 308.6 m ice core to bedrock was recovered from the Guliya ice cap located at 35°17'N, 81°29'E on the northwest Tibetan Plateau (Figure 1). The drilling site is at an elevation of 6200 m a.s.l. Top 266 m of the core was dated to a period spanning 110 ka, and ice below 290 m depth was suggested to be more than 500 ka old due to $^{36}$Cl-dead in the ice (Thompson et al., 1997). Three Guliya interstadials (Stages 3, 5a,

and 5c) are marked by increases in $\delta^{18}O$ values similar to that of the Holocene and Eemian (~124 ka ago) (Thompson et al., 1997).

The Kesang Cave is located in the Tekesi County, western China (42°52′ N, 81°45′ E, elevation ~2000 m a.s.l.) (Figure 1). Eight samples from the Kesang Cave were collected to establish the Kesang $\delta^{18}O$ record with three covering the Holocene and five covering the rest of the Pleistocene portion. Cheng et al. (2012) obtained precise ages (~150 dates), all in stratigraphic order within errors, using a $^{230}Th$ dating technique in the University of Minnesota. The stalagmite $\delta^{18}O$ variations largely reflect changes in the $\delta^{18}O$ of meteoric precipitation (Cheng et al., 2012).

To reconcile the difference in the $\delta^{18}O$ variations between the Guliya and the Kesang records, Cheng et al. (2012) suggested that the Guliya record needs to be younger about a factor of two.

2) It seems that the dating has been performed on basal ice, however it is a little unclear and should be more clear!

Yes the dating was performed on the basal sediment. In fact, this sediment is a mixture of particles and ice. We will clarify this content in the revision.

3) "Ice content" and "water content" seem to be used randomly. This should be more clear.

We took a small portion of the sediment (13.4 g) for measuring ice content, which is determined by weighting the mass before and after drying, resulting in ~30% ice (water equivalent) content.

4) The influence of the ice matrix on the dose-rate should be accounted for in detail and explained and an evaluation of dose-rate for each scenario should be performed.

The infinite matrix dose rate was estimated using concentration-to-dose rate conversion constants presented by Adamiec and Aitken (1998) and the estimate of the dilution of the external dose rate by ice was assumed to be consistent with calculations recommended by Aitken (1985). It's well recognized that water content is one of the major uncertainty factors of the Luminescence dose rate calculation because we do not know the real situation of sediments during geological history. To approach this, two extreme situations for the sediment were presumed, one is zero water content which represents no ice (or water), and the other is the sediment full immersed by the water (saturation water content), i.e., $30 \pm 15\%$ water content in our case. Under these two situations, the maximum and minimum dose rates from water content were estimated. For these uncertainties, we give only an upper age constraint of $42 \pm 4$ ka.

Adamiec, G. and Aitken, M.J.: Dose-rate conversion factors: update, Anc. TL, 16, 37-50, 1998.

Aitken, M. J.: Thermoluminescence dating, Academic Press, London. 1985.

5) The photograph of the Core 1 show a very clear transition to basal ice in the core, however Core 4+5 are retrieved at a different place at the ice-cap where the contourlines in the map of Figure S1 suggest more ice dynamics, and the bottom part of the cores can be much more mixed. The 4-11 micro-metres fin-grained quartz used for the dating could be eolean material deposited onto the ice and therefore younger than the ice-cap. If this is the case, the grains would recieve most of their dose after mixing with the basal ice. The authors should discuss this possibillity.

From Fig. S2, we notice that fast component of the fine grain aliquots accounts for around 87% of the natural OSL signal, indicating that the signal is fast component

dominant. If for a high bedrock fraction, medium and slow components should be high due to no or less bleaching of the bedrock. So the fine quartz grains used for the dating are mostly an eolian origin. The OSL age of the eolian component would represent the time for the ice to move from the surface to the bed, which is younger than the ice cap. For this specific reason, as well as other uncertainties, our result can be only regarded as an upper age constraint.

[Figure]

Figure S2: Natural OSL decay curves and their relative components for the fine grain quartz aliquots. Sum, F, M and S represent natural OSL signal, fast, medium and slow components, respectively. Fitting curve is matched with "Luminescence Analyst" program.

6) In the conclusion the authors suggest collecting more suitable glacier basal sediment. It should be explained what "suitable" means.

We have drilled ice cores from several glaciers and ice caps on the Tibetan Plateau. This is the first time to collect sufficient amount of sediment for the luminescence dating. To avoid misunderstanding, we revised this sentence as the following.

The major limitation of the current work is the very small number of absolute datings and simple assumptions, but this preliminary work provides potential implications for exploring age of mountainous glacier bottom ice. Future work should include collecting more glacier basal sediment samples for the luminescence dating, better understanding the unique processes for preserving the luminescence signal in the glacier basal sediment, and effect of ice on for the dose rate.

Technical corrections:

page 1, line 9: more than one order of magnitude younger

Revised accordingly.

page 1, line 15: interpretation of this information.

Revised accordingly.

page 5, line 5-6: The sentence "We have no information about the behavior of ice in the sediment" should be refrased.

We revised this sentence as the following.

It's well recognized that water content is one of the major uncertainty factors of the Luminescence dose rate calculation because we do not know the real situation of sediments during geological history. To approach this, two extreme situations for the sediment were presumed, one is zero water content which represents no ice (or water),

and the other is the sediment full immersed by the water (saturation water content), i.e., 30 ± 15% water content in our case.

Dear J. Gombiner,

Many thanks for the constructive comments. Below I have made a point-to-point response to the comments. The comments are in black, and our response is in blue.

Sincerely yours,

Hou Shugui
* * *
This is an interesting paper that should eventually be published.

However, the authors could give more thought to the calculation of dose rate and the meaning of the optical age for basal sediment.

The OSL age is the equivalent dose divided by the dose rate. In calculating the OSL age, the authors calculated a lower limit age for dehydrated sediment, containing air in the pore space, and an upper limit age, for hydrated sediment containing water in the pore space. The dose rate is lower for hydrated sediment because water attenuates radiation transfer from grain to grain. The actual sample came from sediment embedded within ice. The authors should calculate a dose rate for the real situation of sediment in ice.

There are potentially two additional sources of radiation that are not included in the dose rate calculation.

(1) Radiation from the bedrock or subglacial sediment.

(2) Radiation from dust layers in the core.

The authors should add these sources of radiation to the dose rate, or show that they are insignificant. If these other sources are included, the higher radiation dose rate would lower the calculated age.

Yes we fully agree that the dose rate is determined by many factors, including the potentially two additional sources of radiation as indicated above.

At first, I need to justify that the sediment samples that we measured for luminescence dating was collected from the very bottom section of the Chongce 216.6 m ice core. Its high particle content (~70%) suggests a similar condition as shown by the inset photo of Figure S1. This is exactly the "subglacial sediment" mentioned here.

[Figure]

Figure S1: Map showing part of the Chongce ice cap where our ice cores were recovered. Core 1 (133.8 m) and Core 2 (135.8 m) to bedrock and Core 3 (58.8 m) were drilled at an altitude of 6010 m a.s.l. in 2012. Core 4 (216.6 m) and Core 5 (208.6 m) to bedrock were drilled at an altitude of 6100 m a.s.l. in 2013. The sediment samples that we measured for luminescence dating was collected from the very bottom section of Core 4, which is similar to the bottom several centimeters section as shown by the inset photo of Core 2.

As to (2) Radiation from dust layers in the core. If the "dust layers" here means the

dust in the bottom section, it's what we measured for luminescence dating. If the "dust layers" here means the dust layers that are frequently observed along the other section of Core 4, as shown by the photo below, they are much weaker than the bottom section. The dust concentration in this kind of dust layers is usually ~3 orders of magnitude lower than the bottom sediment section (Li et al., 2006), suggesting insignificant influence of radiation from dust layers in the core, given a similar radiation intensity of the dust layers in the core and in the bottom sediment.

[Figure]

In fact, we have, so far, no evidence about radiation from the bedrock. We notice that Willerslev et al. (2007) provided the first luminescence measurements on the single grains of quartz and feldspar extracted from a sample cut out of an opaque part of the Greenland Dye 3 basal ice containing dispersed sandy and silty particles. They found that the dose rate contribution from the underlying bedrock was negligible following calculations based on radiation transport modelling software (MCNP5). We are not able to calculate the dose rate contribution from the underlying bedrock at our drilling site due to lack of observation results of the underlying bedrock. For this reason, the dose rate we adopted to calculate the age might be smaller than the real one. So we only provide an upper constraint for the age of the sediment.

Li Y., Yang Y., Han J., Xie Z., M. Nakawo, K. Goto-Azuma. Persistent decrease of dust burden for about 100 years over middle-upper Troposphere of the southern Taklimakan Desert , China. J. Glaciol. Geocryol., 28, 873-878, 2006. (in

Chinese with English abstract)

Willerslev, E., Cappellini, E., Boomsma, W., Nielsen, R., Hebsgaard, M. B., Brand, T. B., Hofreiter, M., Bunce, M., Poinar, H. N., Dahl-Jensen, D., Johnsen, S., Steffensen, J. P., Bennike, O., Schwenninger, J.-L., Nathan, R., Armitage, S., de Hoog, C.-J., Alfimov, V., Christl, M., Beer, J., Muscheler, R., Barker, J., Sharp, M., Penkman, K. E. H., Haile, J., Taberlet, P., Gilbert, M. T. P., Casoli, A., Campani, E., and Collins, M. J.: Ancient biomolecules from deep ice cores reveal a forested southern Greenland, Science, 317, 111-114, doi: 10.1126/science.1141758, 2007.

Finally, I am not sure that the OSL age of the basal sediment directly relates to the age of the ice cap. The authors suggest that the sand-sized quartz grains are sourced from subglacial erosion. If true, it seems likely that some of the silt-sized quartz is also derived from subglacial erosion. Thus, it is conceivable that the dated aliquots are a mixture of eolian quartz and subglacially derived quartz.

Sharp and Gomez (1986) suggested that quartz particles should be "tools" producing comminution of softer minerals within subglacial environments while they themselves remain unaltered. We have noticed the work by Wright (1995), who performed glacial grinding simulations by using a ring shear box designed to examine abrasion and size reduction of quartz grains under "grinding" conditions analogous to those experienced in a subglacial environment. The findings from experimental runs suggest that, although particle breakage and comminution does occur, little silt-sized material is actually produced. The amount of silt produced ranges from approximately 9% of the original sample in one experimental run to less than 1% for the other seven runs.

Our direct evidence from Fig. S2, we notice that fast component of the fine grain aliquots accounts for around 87% of the natural OSL signal, indicating that the signal

is fast component dominant. If for a high bedrock fraction, medium and slow components should be high due to no or less bleaching of the bedrock. So the fine quartz grains used for the dating are mostly an eolian origin.

[Figure]

Figure S2: Natural OSL decay curves and their relative components for the fine grain quartz aliquots. Sum, F, M and S represent natural OSL signal, fast, medium and slow components, respectively. Fitting curve is matched with "Luminescence Analyst" program.

Sharp, M. and Gomez, B.: Processes of debris comminution in the glacial environment and implications for quartz sand-grain micromorphology, Sediment. Geol., 46, 33-47, 1986.

Wright, J. S.: Glacial comminution of quartz sand grains and the production of loessic

silt: a simulation study, Quat. Sci. Rev., 14, 669-680, 1995.

If the ice flow at the core site is dominated by downward vertical motion, then the OSL age of the eolian component of the dated aliquots would represent the time for the ice to move from the surface to the bed, not the age of the ice cap itself.

Yes we agree with the comment, and this will be clarified in the revision.

[revised manuscript text omitted]

---

## Author Response (AR3)

Dear Editor,

Many thanks for the further constructive comments from you and the reviewer. Below I have made a point-to-point response to the comments. The comments are in black, and our response is in blue. I have revised the original manuscript following the comments.

In your previous comments, you suggested that:
- Table 1 provides results from sample CCICE -please clarify the abbreviation. Where do you provide the information on the 8 aliquots?

We previously planed to archive the information in a public database. Now I think it better to include information on all the aliquots in the Supplement, for the convenience of the readers.

Sincerely yours,

Hou Shugui
* * *
Dear authors,

One reviewer has commented your revised paper again. Please address all comments and supply a point to point reply, as well as a revised version of your paper using tracked changes mode.

Since there remain open question about the OSL dating on the basal sediment of the Chongce ice cap (determination of dose rate, identification of the sediment origin), the uncertainties of the resulting ages should be clearly and openly stated and discussed. Addressing these uncertainties in this pioneer work is important for future studies.

Yes, we include the following sentences in the revision to clarify the uncertainty.

The major limitation of the current work is the very small number of absolute ages and simple assumptions, and there exists much uncertainty, such as determination of the dose rate, additional radiation sources from bedrock and dust layers in the core, specific processes underneath the glacier, and so on. For these reasons, we only suggest an upper constraint for the age of the sediment.
* * *
Response to Anonymous Referee #3

The authors present the first luminescence dating of an Tibetan glacier basal sediment and aim to put their findings in a regional context. They have answered to most of the reviewers and the editors questions and comments and changed the manuscript accordingly. The experimental design and technical procedures (drilling, luminescence sample preparation and measurements) are well described and embedded in the research context.

Nevertheless the general concern remains (as also mentioned by the authors itselves) that only one luminescence age of a methodically difficult environment is a weak

basis to draw conclusions concerning the development of the Chongce ice cap.

Specific comments:

- The coarse grain quartz measurements were not considered for age calculation. The authors argue that coarse grained quartz did not match the quality criteria for a reliable dating. They should test whether the slow trend in the OSL decay curve is really due to a dominance of a medium or slow signal component of the quartz? This curve shape could also be attributed to a signal contribution of the feldspar even after IR stimulation according the double SAR protocol of Banerjee et al. (depending on the pre-OSL IR stimulation temperature). If medium or slow components were not clearly detected, why not presenting the coarse grain quartz result in the paper (as the authors did answering the referees comments) and compare them with the fine grain dating result? Coarse grain ages are slightly older compared to the fine grain, but they are in a similar age range which would be interesting to state.

Following this comment, we provide coarse grain ages in the Supplement.

Table S3. Results of the coarse quartz grains with their corresponding OSL ages.

| Sample | U (ppm) | | Th (ppm) | | K (%) | | $D_e$ | Water content | Dose rate | Age |
|---|---|---|---|---|---|---|---|---|---|---|
| | Gamma | NAA | Gamma | NAA | Gamma | NAA | (Gy) | (%) | (Gy/ka) | (ka) |
| CCICE* | 3.66 ± 0.15 | 3.45 ± 0.12 | 11.21 ± 0.42 | 11.40 ± 0.32 | 3.52 ± 0.10 | 3.48 ± 0.08 | 238 ± 51 | 0 | 5.25 ± 0.45 | 45 ± 11 |
| | | | | | | | | 30 ± 15 | 3.85 ± 0.24 | 62 ± 14 |

* CCICE stands for Chongce Ice.

- Page 4, lines 25, 26: In case of fine grain OSL dating, a narrow and symmetric De distribution does not necessarily indicate sufficient bleaching due to averaging effects.

Following this comment, we have revised accordingly as:

In addition, the $D_e$ distribution presented in Fig. 3 shows a narrow and nearly symmetric shape for the fine quartz grains, possibly indicating sufficient bleaching of the fine quartz grains. These results imply that the fine grain quartz might be more appropriate for luminescence dating than the coarse grain quartz. The same conclusion was also reached in some previous studies on Himalayan glacial deposits (e.g. Hu et al. 2015). Thus we prefer to the results of the fine grains, but provide the coarse grain quartz results in the Supplement (Table S3). It's apparent that even the coarse grain ages are slightly older compared to the fine grain ages (Table 1), they are in a similar age range.

- It is always unclear to me which event is dated by OSL in this study. Sure it is the last daylight exposure of the mineral grains, but where the material was bleached? Different scenarios are possible depending on the origin of the dated material and how it came to the present position at the base of the ice cap. Depending on the postulated sedimentation process, the OSL ages date more or less the formation of the ice cap, but they can also predate or postdate this process.

I suggest to better work out what the OSL age implies for the timing of the ice cap formation and to provide different scenarios to structure the way of argumentation.

We agree with all the comments, but with only one upper age constraint of the current work, we think it better not to go much further. In the manuscript, we clarified the major limitation and uncertainty concerning to the current work. Apparently, much more work should be required to decipher the different possible scenarios, for instance, to drill ice cores to the bottom at different elevations of the glaciers for their bottom ages. We are looking forward to more new results.

In the revision, we include the following sentences to clarify the uncertainty.

The major limitation of the current work is the very small number of absolute ages and simple assumptions, and there exists much uncertainty, such as determination of the dose rate, additional radiation sources from bedrock and dust layers in the core, specific processes underneath the glacier, and so on. For these reasons, we only suggest an upper constraint for the age of the sediment.

- It should be clarified why the OSL age implies ice-free conditions below the elevation of the bottom at the drilling site in warmer periods later than 42 ± 4 ka (Bölling, Alleröd, Holocene).

If in the cold (glacial) periods later than 42 ± 4 ka, the Chongce ice cap should be larger than its present area. So only in warmer periods later than 42 ± 4 ka, the Chongce ice cap could be smaller than its present area, implying ice-free conditions below the elevation of the bottom at the drilling site.

Technical comment:

- Page 3, line 24: Please correct "Stocks' law" to "Stoke's law".

We apologize for this negligence. Have made correction accordingly.

[revised manuscript text omitted]

---

## Author Response (AR4)

Dear Editor,

Many thanks for your positive opinion and further comments on corrections of the English language. Below I have made a point-to-point response to the comments. The comments are in black, and our response is in blue. I have revised the original manuscript following the comments.

Sincerely yours,

Hou Shugui
* * *
Comments to the Author:

Dear authors,

Thanks for carefully considering all suggestions by the reviewers and providing further explanations. I am pleased to accept your paper for publication in TC after some minor technical changes, which include corrections of the English language. Specifically, the added sentences need English clarification:

Page 4, Line 29: Thus we prefer to the results of the fine grains, but provide the coarse grain quartz results in the Supplement (Table S3).

Should it be "We therefore used" and "results from the fine quartz grains"?

We have changed this sentence as:

We therefore used the results of the fine grains, and the results from the coarse grain quartz were given in the Supplement (Table S3).

Page 4, Line 30: It's apparent that even the coarse grain ages are slightly older compared to the fine grain ages (Table 1), they are in a similar age range.

My suggestion for change:

Although the coarse grain ages are slightly older than the fine grain ages (Table 1), they clearly fall within a similar range.

Yes we have changed this sentence as:

Although the coarse grain ages are slightly older than the fine grain ages (Table 1), they clearly fall within a similar range.